# A Simple Baseline for Task-Agnostic Self-Training

## Abstract

Classical semi-supervised approaches achieve state-of-the-art results on various visual recognition tasks, especially image classification, but they are typically designed with expert knowledge of the task at hand such as task-specific data augmentation. However, these approaches do not generalize to novel tasks such as image segmentation and surface normal estimation. In this work, we instead study self-training for a wide variety of tasks in a task-agnostic fashion. We find out a simple success recipe: to construct a continuous schedule of learning updates that iterates between self-training on novel segments of the streams of unlabeled data, and fine-tuning on the small and fixed labeled data. Our task-agnostic self-training approach works with a few labeled samples per task by leveraging millions of unlabeled web images, and it requires neither enormous computational resources to process data nor domain-specific unlabeled data, which are assumed in most prior works. We show that our simple approach, without hyper-parameter tuning, can be as effective as state-of-the-art semi-supervised learning method (Fixmatch) that is designed with task-specific knowledge for image classification. Furthermore, we demonstrate the findings for both (1) pixel-level tasks such as surface normal estimation and segmentation, and (2) diverse domains with extreme differences to web images, including medical, satellite, and agricultural imagery, where there does not exist a large amount of labeled or unlabeled data. The experiments consistently suggest that ours is a competitive baseline to consider before developing compute-heavy and task-specific semi-supervised methods.

## 1 Introduction

Training a visual recognition model requires enormous domain-specific resources, specifically (1) large amount of high-quality curated labeled data (Krizhevsky et al., 2012; Lin et al., 2014); (2) extensive computational resources (Radford et al., 2021; Devlin et al., 2015) (disk space to store data and GPUs to process it); (3) task-specific optimization or dataset-specific knowledge to tune hyperparameters (Sohn et al., 2020; Berthelot et al., 2019; Cai et al., 2022; Xu et al., 2022). In this work, we study the role of domain-agnostic unlabeled images to improve a visual recognition model in a task-agnostic fashion. Recent semi-supervised approaches (Xie et al., 2020b; Yalniz et al., 2019; Cai et al., 2022) (not requiring extensive labeled data) may cost a million dollar budget for AWS compute resources. Our goal is to benefit the advances in semi-supervised learning with minimal resources. We present a simple success recipe for task-agnostic self-training which allows a user to train a visual recognition model from a few labeled examples (Fig. 1-(a)) and a domain-agnostic streams of unlabeled web images (Fig. 1-(b)).

**Self-Training and Semi-Supervised Learning:** A large variety of self-training (Du et al., 2020; Wei et al., 2020) and semi-supervised approaches (Radosavovic et al., 2018; Scudder, 1965; Van Engelen & Hoos, 2020; Yarowsky, 1995; Xie et al., 2020b; Yalniz et al., 2019) use unlabeled images in conjunction with labeled images to learn a better representation (Fig. 2-(b)). These approaches require: (1) task-specific knowledge such as better loss-functions and augmentation tricks for image classification tasks (Azuri & Weinshall, 2020; Barz & Denzler, 2020; Cai et al., 2022); (2) intensive computational requirements (Radosavovic et al., 2018; Xie et al., 2020b; Yalniz et al., 2019) or heavy hyperparameter tuning such as thresholding noisy pseudo-labels (Arazo et al., 2020; Iscen et al., 2019; Lee, 2013; Xie et al., 2020b; Yalniz et al., 2019); and (3) a large domain-specific unlabeled dataset sampled from same or similar data distribution as that of labeled

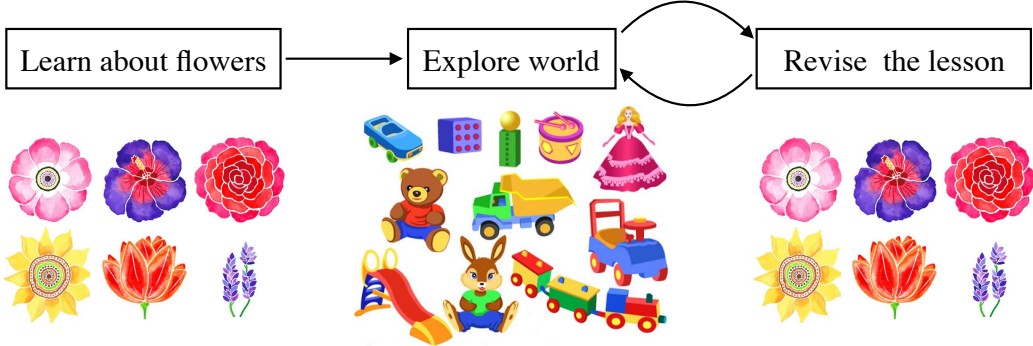

(a) Children continually improve their knowledge about a concept.

Flowers-102 dataset: 10 examples per class

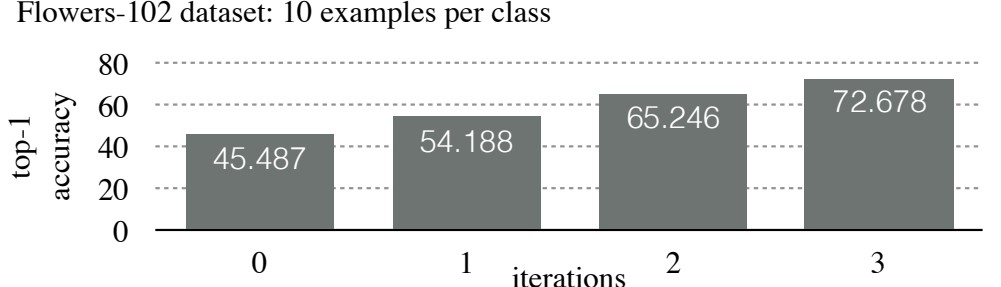

(b) Machines can also improve their knowledge about a concept in this iterative manner.

Figure 1: **(a)** We take inspiration from developmental psychology that explores howchildren learn. Children maybe exposed to a concept (say flowers), play with other things in their environment, and eventually return to the lesson at hand. By interleaving periods of self-supervised play and teacher-supervised learning, they can continually evolve their representations about the world. **(b)** We use unlabeled web images to improve the performance for various tasks without any task-specific or domain-specific knowledge.

examples (Berthelot et al., 2020; 2019; Chen et al., 2018; Iscen et al., 2019; Lerner et al., 2020; Phoo & Hariharan, 2021; Sohn et al., 2020; Xie et al., 2020a; Xu et al., 2022). We differ from this setup. In this work, the unlabeled data is domain-agnostic and have no relation with the intended task. We use a 4 GPU (GeForce RTX 2080) machine to conduct all our experiments. Finally, we do not apply any advanced optimization schema, neither we apply any task-specific knowledge nor we tune any hyperparameters.

**Streams of Unlabeled Web Images:** Existing semi-supervised methods use unlabeled data from similar data distribution (Berthelot et al., 2020; 2019; Chen et al., 2018; Iscen et al., 2019; Lerner et al., 2020; Phoo & Hariharan, 2021; Sohn et al., 2020; Xie et al., 2020a). In this work, we observe that unlabeled examples from quite different data distributions can still be helpful. We make use of domain-agnostic unlabeled streams of web images (such as ImageNet-21K (Deng et al., 2009), YFCC100M (Thomee et al., 2016), and INaturalist[1] (Van Horn et al., 2018)) to improve a variety of domain-specific tasks defined on satellite images, agricultural images, and even medical images. Starting from a very few labeled examples, we iteratively improve task performance by constructing a schedule of learning updates that iterates between pre-training on segments of the unlabeled stream and fine-tuning on the small labeled dataset (Fig. 2-(e)). We progressively learn more accurate pseudo-labels as the stream is processed. This observation implies that we can learn better mappings using diverse unlabeled examples without any extra supervision or knowledge of the task.

---

[1] 2021's version at: `https://github.com/visipedia/inat_comp/tree/master/2021`

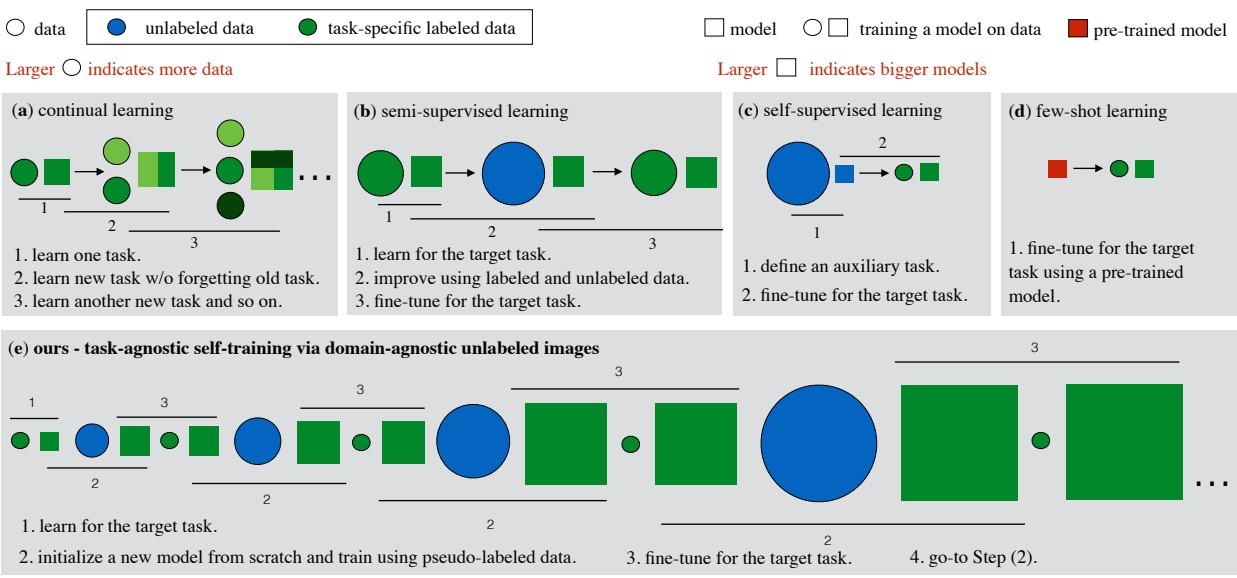

Figure 2: **Task-Agnostic Self-Training and Established Methods:** (a) **Continual learning** continually learns new tasks in a supervised manner without forgetting previous ones. Our approach can continuously learn better models for a fixed task using an infinite stream of unlabeled data.(b) **Semi-supervised learning** typically requires (1) a large domain-specific unlabeled dataset sampled from same or similar data distribution as that of labeled examples (Berthelot et al., 2020; 2019; Chen et al., 2018; Iscen et al., 2019; Lerner et al., 2020; Phoo & Hariharan, 2021; Sohn et al., 2020; Xie et al., 2020a); (2) intensive computational resources (Radosavovic et al., 2018; Xie et al., 2020b; Yalniz et al., 2019); and (3) task-specific knowledge such as better loss-functions for image classification tasks (Azuri & Weinshall, 2020; Barz & Denzler, 2020) or cleaning noisy pseudo-labels (Arazo et al., 2020; Iscen et al., 2019; Lee, 2013; Xie et al., 2020b; Yalniz et al., 2019). In contrast, our approach makes use of unlabeled data that is domain-agnostic and has no relation with the intended task. We also require modest compute; we use a 4 GPU (GeForce RTX 2080) machine to conduct all our experiments. (c) **Self-supervised learning** methods learn a representation from unlabeled images using an auxiliary task. This learned representation can then be fine-tuned for downstream target task. In this work, we explore the role of unlabeled images for the target task without defining an auxiliary task. Our work shares insights with Chen et al. (2020) that use big self-supervised models for semi-supervised learning. We find it to be true even when using impoverished models for initialization, i.e., training the model from scratch for a task given a few labeled examples. The performance for the task is improved over time in a streaming/iterative manner. While we do observe the benefits of having a better initialization, we initialize the models from scratch for a task throughout this work. (d) **Few-shot learning** learns representations from a few-labeled examples. Guo et al. (Guo et al., 2020) show that popular few-shot learning methods (Finn et al., 2017; Lee et al., 2019; Snell et al., 2017; Sung et al., 2018; Tseng et al., 2020; Vinyals et al., 2016) underperform simple finetuning, i.e., when a model pre-trained on large annotated datasets from similar domains is used as an initialization to the few-shot target task. The subsequent tasks in few-shot learners are often tied to both original data distribution and tasks. Our approach makes use of few-labeled examples but it is both task-agnostic and domain-agnostic.

**Our Contributions:** (1) We use the same method for a wide variety of tasks using the same set of unlabeled images. In this work, we present a study to empirically understand the role of domain-agnostic unlabeled web images to learn a better representation without any task-specific knowledge. We demonstrate this behaviour for the tasks where data distribution of unlabeled images drastically varies from the labeled examples of the intended tasks, such as using web images for medical-image classification, crop-disease classification, and satellite-image classification. We improve surface normal estimation on NYU-v2 depth dataset (Silberman et al., 2012) and semantic segmentation on PASCAL VOC-2012 (Everingham et al., 2010) by $3-7\%$; (2) We observe that one can improve the performance by leveraging more streams of unlabeled data on fine-grained image classification tasks. Without any domain-specific or task-specific knowledge, we improve the results in few iterations of our approach. We observe that learning can be made faster by increasing the capacity of models; and (3) finally, we study that how these insights allow us to design an efficient and cost-effective system for a non-expert.

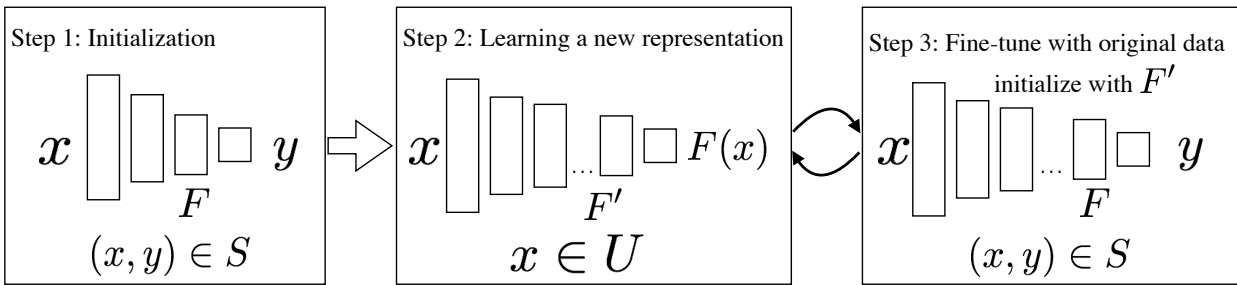

Figure 3: **Our Approach:** There are three important steps of our approach. (a) **Step 1:** Initialization– we learn an initial mapping $F$ on $(x, y) \in S$; (b) **Step 2:** Learning a new representation– We use $F$ to learn a new model $F'$ from scratch on sample $x \in U$; and (c) finally, **Step 3:** Fine-tune with original data – we fine-tune $F'$ on $S$. This becomes our new $F$. We continually cycle between Step-2 and Step-3. The capacity of model $F'$ increases with every cycle.

## 2 Related Work

Our work is inspired from the continuously improving and expanding human mind (Ahn & Brewer, 1993; Ahn et al., 1987). Prior work focuses on one-stage approaches for learning representations for a task, typically via more labeled data (Lin et al., 2014; Russakovsky et al., 2015; Zhou et al., 2017), higher capacity parametric models (He et al., 2016; Huang et al., 2017; Krizhevsky et al., 2012; Simonyan & Zisserman, 2015), finding better architectures (Cao et al., 2019; Tan & Le, 2019; Zoph et al., 2018), or adding task-specific expert knowledge to train better models (Qi et al., 2018; Wang et al., 2015).

**Continual and Iterated Learning:** Our work shares inspiration with a large body of work on continual and lifelong learning (Thrun, 1996; 1998; Silver et al., 2013). A major goal in this line of work (Finn et al., 2017; 2019; Rao et al., 2019; Rebuffi et al., 2017; Wallingford et al., 2020) has been to continually learn a good representation over a sequence of tasks (Fig. 2-(a)) that can be used to adapt to a new task with few-labeled examples without forgetting the earlier tasks (Castro et al., 2018; Li & Hoiem, 2017). Our goal, however, is to learn better models for a task given a few labeled examples without any extra knowledge. Our work shares insights with iterated learning (Kirby, 2001; Kirby et al., 2014) that suggests evolution of language and emerging compositional structure of human language through the successive re-learning. Recent work (Lu et al., 2020b;a) has also used these insights in countering language drift and interactive language learning. In this work, we restrict ourselves to visual recognition tasks and show that we can get better task performance in an iterated learning fashion using infinite stream of unlabeled data.

**Learning from Unlabeled or Weakly-Labeled Data:** The power of large corpus of unlabeled or weakly-labeled data has been widely explored in semi-supervised learning (Arazo et al., 2020; Chapelle et al., 2009; Iscen et al., 2019; Nigam et al., 2000; Radford et al., 2018; Radosavovic et al., 2018; Raina et al., 2007; Zhang et al., 2016b; Zhu, 2005), self-supervised learning (Fig. 2-(c)) (Doersch et al., 2015; Gidaris et al., 2018; Zhang et al., 2016a), or weakly-supervised learning (Izadinia et al., 2015; Joulin et al., 2016; Sun et al., 2017; Zhou, 2018). Self-supervised approaches learn a representation from unlabeled images via an auxiliary task. The learned model is then fine-tuned for the target task. In this work, we explore the use of domain-agnostic unlabeled examples to learn a representation for the target task without any auxiliary task. A wide variety of work in few-shot learning (Li et al., 2019; Ravi & Larochelle, 2017; Wang et al., 2018; Wertheimer & Hariharan, 2019), meta-learning (Ren et al., 2018; Snell et al., 2017; Sung et al., 2018) aims to learn from few labeled samples. These approaches largely aim at learning a better generic visual representation from a few labeled examples (Fig. 2-(d)). In this work, we too use few labeled samples for the task of interest along with large amounts of domain-agnostic unlabeled images. Our goal is to learn a better model for any task without any domain biases, neither employing extensive computational resources nor expert human resources. Our work shares insights with Chen et al. (2020) that use big self-supervised models for semi-supervised learning. We observe that it is true even when using impoverished models for initialization, i.e., training the model from scratch for a task given a few labeled examples. The performance

for the task is improved over time in a streaming/iterative manner. While we do observe the benefits of having a better initialization (Sec 4.1.3), we initialize the models from scratch for a task for all our analysis throughout this work.

**Domain Biases and Agnosticism:** Guo et al. (Guo et al., 2020) show that meta-learning methods (Finn et al., 2017; Lee et al., 2019; Snell et al., 2017; Sung et al., 2018; Tseng et al., 2020; Vinyals et al., 2016) underperform simple finetuning, i.e., when a model pre-trained on large annotated datasets from similar domains is used as an initialization to the few-shot target task. The subsequent tasks in few-shot learners are often tied to both original data distribution and tasks. Our approach makes use of few-labeled examples but it is both task-agnostic and domain-agnostic. In this work, we initialize models from scratch (random gaussian initialization) from a few labeled examples. In many cases, we observe that training from scratch with a few-labeled examples already competes with fine-tuning a model pretrained on large labeled dataset. Specifically, we show substantial performance improvement in surface normal estimation (Fouhey et al., 2013; Wang et al., 2015) on NYU-v2-depth (Silberman et al., 2012) (that is primarily an indoor world dataset collected using a Kinect) via an unlabeled stream of web images. We similarly show that unlabeled Internet streams can be used to improve classification accuracy of crop-diseases (Russakovsky et al., 2015), satellite imagery (Helber et al., 2019), and medical images (Codella et al., 2019; Tschandl et al., 2018) with a modest number of labeled examples (20 examples per class).

**Avoiding Overfitting:** An important consequence of our work is that we can now train very deep models from scratch using a few labeled examples without any expert neural network knowledge. The large capacity models are often prone to overfitting in a low-data regime and usually under-perform (Newell & Deng, 2020). For e.g. a ResNet-50 model (He et al., 2016) trained from scratch (via a softmax loss) for a 200-way fine-grained bird classification (Welinder et al., 2010) using 30 examples-per-class overfits and yields 21.7% top-1 accuracy on a held-out validation set. In a single iteration of our approach, the same model gets 51.5% top-1 accuracy in a day. We take inspiration from prior art on growing networks (Wang et al., 2017; Wen et al., 2016; Zhang & Yu, 2020) that slowly "grow" the network using unlabeled examples from similar distribution. In this work, we observe that we can quickly increase the capacity of model by streaming learning via a large amount of diverse unlabeled images. This is crucial specially when there is a possibility of a better representation but we could not explore them because of the lack of labeled and unlabeled data from similar distribution. Our "growing" mechanism is also much simpler compared to prior arts; we can just replace ResNet18 by any larger capacity models such as ResNet50, resulting in only one line of code change.

## 3 Method

Our streaming learning approach is both an extension and a simplification of state-of-the-arts semi-supervised learning algorithms such as (Yalniz et al., 2019). To derive our approach, assume we have access to an optimization routine that minimizes the loss on a supervised data set of labeled examples $(x, y) \in S$:

$$\text{Learn}(\mathcal{H}, S) \leftarrow \underset{F \in \mathcal{H}}{\arg\min} \sum_{(x,y) \in S} \text{loss}(y, F(x)) \tag{1}$$

We will explore continually-evolving learning paradigms where the model class $\mathcal{H}$ grows in complexity over time (e.g., deeper models). We assume the gradient-based optimization routine is randomly initialized "from scratch" unless otherwise stated.

**Semi-supervised learning:** In practice, labeled samples are often limited. Semi-supervised learning assumes one has access to a large amount of unlabeled data $x \in U$. We specifically build on a family of deep semi-supervised approaches that psuedo-label unsupervised data $U$ with a model trained on supervised data $S$ (Arazo et al., 2020; Iscen et al., 2019; Lee, 2013). Since these psuedo-labels will be noisy, it is common to pre-train on this large set, but fine-tune the final model on the pristine supervised set $S$ (Yalniz et al., 2019). Specifically, after learning an initial model $F$ on the supervised set $S$:

1. Use $F$ to psuedo-label $U$.

---

**Algorithm 1:** StreamLearning($S, \{U_t\}_{t=1}^T, \{\mathcal{H}_t\}_{t=1}^T$)

---

**Input** : $S$: Labeled dataset

$\{U_t\}_{t=1}^T$: $T$ slices from unlabeled stream

$\{\mathcal{H}_t\}_{t=1}^T$: $T$ hypothesis classes

**Output:** $F$

```
// Initialize the model on S
```

$F \leftarrow \text{Learn}(\mathcal{H}_1, S)$;

**for** $t \leftarrow 1$ *to* $T$ **do**

    `// Pseudo-label stream slice`

    $U \leftarrow \{(x, F(x)) : x \in U_t\}$;

    `// Pretrain model on` $U$

    $F' \leftarrow \text{Learn}(\mathcal{H}_t, U)$;

    `// Fine-tune model on` $S$

    $F \leftarrow \text{Finetune}(F', S)$;

**end**

---

2. Learn a new model $F'$ from random initialization on the pseudo-labelled $U$.

3. Fine-tune $F'$ on S.

**Iterative learning:** The above 3 steps can be iterated for improved performance, visually shown in Fig. 3. It is natural to ask whether repeated iteration will potentially oscillate or necessarily converge to a stable model and set of pseudo-labels. The above iterative algorithm can be written as an approximate coordinate descent optimization (Wright, 2015) of a latent-variable objective function:

$$\min_{\{z\}, F \in \mathcal{H}} \sum_{(x,y) \in S} \text{loss}(y, F(x)) + \sum_{x \in U} \text{loss}(z, F(x)) \tag{2}$$

Step 1 optimizes for latent labels $\{z\}$ that minimize the loss, which are obtained by assigning them to the output of model $z := F(x)$ for each unlabeled example $x$. Step 2 and 3 optimize for $F$ in a two-stage fashion. Under the (admittedly strong) assumption that this two-stage optimization finds the globally optimal $F$, the above will converge to a fixed point solution. In practice, we do not observe oscillations and find that model accuracy consistently improves.

**Streaming learning:** We point out two important extensions, motivated by the fact that the unsupervised set $U$ can be massively large, or even an infinite stream (e.g., obtained by an online web crawler). In this case, Step 1 may take an exorbitant amount of time to finish labeling on $U$. Instead, it is convenient to "slice" up $U$ into a streaming collection of unsupervised datasets $U_t$ of manageable (but potentially growing) size, and simply replace $U$ with $U_t$ in Step 1 and 2. One significant benefit of this approach is that as $U_t$ grows in size, we can explore larger and deeper models (since our approach allows us to pre-train on an arbitrarily large dataset $U_t$). In practice, we train a family of models $\mathcal{H}_t$ of increasing capacity on $U_t$. Our final streaming learning algorithm is formalized in Alg. 1.

## 4 Experiments

We first study the role of domain-agnostic unlabeled images in Section 4.1. We specifically study tasks where the data distribution of unlabeled images varies drastically from the labeled examples of the intended task. We then study the role of streaming learning in Section 4.2. We consider the well-studied task of fine-grained image classification here. We observe that one can dramatically improve the performance without using any task-specific knowledge. Finally, we study the importance of streaming learning from the perspective of a non-expert, i.e., cost in terms of time and money.

### 4.1 Role of Domain-Agnostic Unlabeled Images

We first contrast our approach with FixMatch (Sohn et al., 2020) in Section 4.1.1. FixMatch is a recent state-of-the-art semi-supervised learning approach that use unlabeled images from similar distributions as that of the labeled data. We contrast FixMatch with our approach in a setup where data distribution of unlabeled images differ from labeled examples, e.g., the unlabeled stream could be ImageNet-21K (Russakovsky et al., 2015), YFCC100M (Thomee et al., 2016), or INaturalist (Van Horn et al., 2018). We then analyze the role of domain-agnostic unlabeled images to improve task-specific image classification in Section 4.1.2. The data distribution of unlabeled images dramatically differs from the labeled examples in this analysis. Finally, we extend our analysis to pixel-level tasks such as surface-normal estimation and semantic segmentation in Section 4.1.3.

#### 4.1.1 Comparison with FixMatch, Sohn et al. (2020)

We use two fine-grained image classification tasks for this study: (1) Flowers-102 (Nilsback & Zisserman, 2008) with 10 labeled examples per class; and (2) CUB-200 (Welinder et al., 2010) with 30 labeled examples per class. The backbone model used is ResNet-18. We conduct analysis in Table 1 where we use the default hyperparameters from FixMatch (Sohn et al., 2020) for analysis.

In specific, we use SGD optimizer with momentum 0.9 and the default augmentation for all experiments (except that FixMatch during training adopts both a strong and a weak (the default) version of image augmentation, whereas our approach only uses the default augmentation). For FixMatch, we train using lr 0.03, a cosine learning rate scheduling, L2 weight decay 5e-4, batch size 256 (with labeled to unlabeled ratio being 1:7) on 4 GPUs with a total of 80400 iterations. For our approach, we first train from scratch only on the labeled samples with the same set of hyperparameters as in FixMatch (with all 256 samples in the batch being labeled samples). From there we could already see that FixMatch sometimes does not match this naive training strategy. Then for our StreamLearning approach, we generate the pseudo-labels on the unlabeled set $U_1$ and trained for another 80400 iterations with lr 0.1 (decay to 0.01 at 67000 iteration), L2 weight decay 1e-4, batch size 256 on 4 GPUs. Finally, we finetuned on the labeled samples for another 80400 iterations with lr 0.1 (decay to 0.01 at 67000 iteration), L2 weight decay 1e-4, batch size 256 on 4 GPUs.

Comparison with FixMatch

| Task | scratch | U = ImageNet | | U = YFCC100M | | U = INat2021 | |
|------|---------|--------------|---------|--------------|---------|--------------|---------|
| | | FixMatch | $U_1$ (ours) | FixMatch | $U_1$ (ours) | FixMatch | $U_1$ (ours) |
| Flowers-102 (Nilsback & Zisserman, 2008) | 58.21 | 53.00 | **61.51** | 54.64 | **63.88** | 54.63 | **60.35** |
| CUB-200 (Welinder et al., 2010) | 44.24 | 51.24 | **60.58** | 49.19 | **59.77** | 45.91 | **54.69** |

Table 1: We contrast our approach with FixMatch (Sohn et al., 2020) on two fine-grained image classification tasks. For unlabeled images, we use a million unlabeled images from either ImageNet (Russakovsky et al., 2015) or YFCC100M (Thomee et al., 2016) for this experiment. The backbone model used is ResNet-18. Our approach significantly outperforms FixMatch. We use the default hyperparameters from FixMatch (Sohn et al., 2020).

#### 4.1.2 Extreme-Task Differences

We use: (1) **EuroSat** (Helber et al., 2019) (satellite imagery) dataset for classifying satellite-captured images into distinct regions; (2) **ISIC2018** (Codella et al., 2019) (lesion diagnosis) for medical-image classification of skin diseases; and (3) **CropDiseases** (Mohanty et al., 2016) dataset which is a crop-disease classification task. We use 20 examples per class for each dataset and train the models from scratch. We provide details about the dataset and training procedure in the Appendix A.1.

Table 2 shows the performance for the three different tasks. We achieve significant improvement for each of them. We also show the performance of a pre-trained (using 1.2M labeled examples from ImageNet) model on these datasets. Guo et al. (2020) suggested that fine-tuning a pre-trained model generally leads

| Task | pre-trained | scratch | $U_1$ (ours) | | |
|------|-------------|---------|----------|---------|----------|
| | | | ImageNet | YFCC100M | INat2021 |
| East-SAT (Helber et al., 2019) | 68.93 | 70.57 | 73.85 | 77.07 | 77.78 |
| Lesion (Codella et al., 2019) | 45.43 | 44.86 | 50.86 | 52.29 | 51.43 |
| Crop (Mohanty et al., 2016) | 94.68 | 87.49 | 90.86 | 91.46 | 90.41 |

Table 2: **Extreme-Task Differences:** We analyse tasks that operate on specialized data distributions. We observe significant performance improvement despite the unlabeled streams of internet images being used (ImageNet or YFCC100M). We also achieve performance competitive to the ImageNet-1k pre-trained model (again, trained with a large amount of labels). We use ResNet-18 for all experiments in the table.

to best performances on these tasks. We observe that a simple random-gaussian initialization works as well despite trained using only a few labeled examples. Crucially, we use unlabeled Internet images for learning a better representation on classification tasks containing classes that are extremely different to real-world object categories. Still, we see significant improvements.

### 4.1.3 Pixel Analysis

We extend our analysis to pixel-level prediction problems. We study surface-normal estimation using NYU-v2 depth dataset (Silberman et al., 2012). We intentionally chose this task because there is a large domain gap between NYU-v2 depth dataset and internet images of ImageNet-21k. We follow the setup of Bansal et al. (Bansal et al., 2017; 2016) for surface normal estimation because: (1) they demonstrate training a reasonable model from scratch; and (2) use the learned representation for downstream tasks. This allows us to do a proper comparison with an established baseline and study the robustness of the models. Finally, it allows us to verify if our approach holds for a different backbone-architecture (VGG-16 (Simonyan & Zisserman, 2015) in this case).

**Evaluation:** We use 654 images from the test set of NYU-v2 depth dataset for evaluation. Following Bansal et al. (2016), we compute six statistics over the angular error between the predicted normals and depth-based normals to evaluate the performance – **Mean**, **Median**, **RMSE**, **11.25°**, **22.5°**, and **30°** – The first three criteria capture the mean, median, and RMSE of angular error, where lower is better. The last three criteria capture the percentage of pixels within a given angular error, where higher is better.

Table 3 contrasts the performance of our approach with Bansal et al. (2016; 2017). They use a pre-trained ImageNet classification model for initialization. In this work, we initialize a model from random gaussian initialization (also known as scratch). The second-last row shows the performance when a model is trained from scratch. We improve this model using a million unlabeled images. The last row shows the performance after one iteration of our approach. We improve by 3-6% without any knowledge of surface normal estimation task. Importantly, we outperform the pre-trained ImageNet initialization. This suggests that we should not limit ourselves to pre-trained classification models that have access to large labeled datasets. We can design better neural network architectures for a task using our approach.

**Can we capture both local and global information without class-specific information?** One may suspect that a model initialized with the weights of pre-trained ImageNet classification model may capture more local information as the pre-training consists of class labels. Table 4 contrast the performance of two approaches on indoor scene furniture categories such as chair, sofa, and bed. The performance of our model exceeds prior art for local objects as well. This suggests that we can capture both local and global information quite well without class-specific information.

Surface Normal Estimation (NYU Depthv2)

| Approach | Mean ↓ | Median ↓ | RMSE ↓ | 11.25° ↑ | 22.5° ↑ | 30° ↑ |
|---|---|---|---|---|---|---|
| Bansal et al. (2016) | 19.8 | 12.0 | 28.2 | 47.9 | 70.0 | 77.8 |
| Goyal et al. (2019) | 22.4 | 13.1 | - | 44.6 | 67.4 | 75.1 |
| init (scratch) | 21.2 | 13.4 | 29.6 | 44.2 | 66.6 | 75.1 |
| $U_1$ (ours) | **18.7** | **10.8** | **27.2** | **51.3** | **71.9** | **79.3** |

Table 3: We contrast the performance of our approach with Bansal et al. (2016; 2017), which is the state-of-the-art given our setup. They use a pre-trained ImageNet classification model for initialization. In this work, we initialize a model from random gaussian initialization. The third row shows the performance of a scratch-initialized model. We improve this model using one million unlabeled images. The last row shows the performance after one iteration of our approach. We improve by 3-6% without any domain-specific knowledge about the surface normal estimation task. Importantly, we outperform the pre-trained ImageNet initialization. We contrast our method with Goyal et al. (2019) (second-row), which use $100M$ unlabeled images to train a generic representation via jigsaw puzzle (Noroozi & Favaro, 2016) using a ResNet-50 model. Our model trained from scratch competes with their best performing model. This analysis suggests two things: (1) we can design better neural network architecture and does have to limit ourselves to pre-trained classification models; and (2) Our approach can learn better models with two-orders less unlabeled data as compared to Goyal et al. (2019).

Per-Object Surface Normal Estimation (NYU Depthv2)

| | Mean ↓ | Median ↓ | RMSE ↓ | 11.25° ↑ | 22.5° ↑ | 30° ↑ |
|---|---|---|---|---|---|---|
| **chair** | | | | | | |
| Bansal et al. (2016) | 31.7 | 24.0 | 40.2 | **21.4** | 47.3 | 58.9 |
| $U_1$ (ours) | **31.2** | **23.6** | **39.6** | 21.0 | **47.9** | **59.8** |
| **sofa** | | | | | | |
| Bansal et al. (2016) | 20.6 | 15.7 | 26.7 | 35.5 | 66.8 | 78.2 |
| $U_1$ (ours) | **20.0** | **15.2** | **26.1** | **37.5** | **67.5** | **79.4** |
| **bed** | | | | | | |
| Bansal et al. (2016) | 19.3 | 13.1 | 26.6 | 44.0 | 70.2 | 80.0 |
| $U_1$ (ours) | **18.4** | **12.3** | **25.5** | **46.5** | **72.7** | **81.7** |

Table 4: We contrast the performance of our approach with the model fine-tuned using ImageNet (with class labels) on furniture categories, i.e. chair, sofa, and bed. Our approach outperforms prior art without any class information.

The details of the model and training procedure used in these experiments are available in the Appendix A.2. We have also provided analysis showing that we capture both local and global details without class-specific information.

**Can we improve *scratch* by training longer?** It is natural to ask if we could improve the performance by training a model from scratch for more iterations. Table 5 shows the performance of training the scratch model for longer (until convergence). We observe that we do improve slightly over the model we use. However, this improvement is negligible in comparisons to streaming learning.

**Is it a robust representation?** Bansal et al. (2017) has used the model trained for surface-normal as an initialization for the task of semantic segmentation. We study if a better surface normal estimation means better initialization for semantic segmentation. We use the training images from PASCAL VOC-2012 (Everingham et al., 2010) for semantic segmentation, and additional labels collected on 8498 images by (Hariharan et al., 2011) for this experiment. We evaluate the performance on the test set that required

| Approach | Mean | Median | RMSE | 11.25° | 22.5° | 30° |
|---|---|---|---|---|---|---|
| Bansal et al. (2016) | 19.8 | 12.0 | 28.2 | 47.9 | 70.0 | 77.8 |
| init | 21.2 | 13.4 | 29.6 | 44.2 | 66.6 | 75.1 |
| init (until convergence) | 20.4 | 12.6 | 28.7 | 46.3 | 68.2 | 76.4 |
| $U_1$ (ours) | **18.7** | **10.8** | **27.2** | **51.3** | **71.9** | **79.3** |

Table 5: **Can we improve scratch by training longer?** It is natural to ask if we can get better performance for training longer, crucially for a model trained from scratch. We observe that one can indeed get a slightly better performance by training for a long time. However, this improvement is negligible compared to ours.

submission on PASCAL web server (pas). We report results using the standard metrics of region intersection over union (**IoU**) averaged over classes (higher is better). Refer to Appendix A.3 for details about training.

We show our findings in Table 6. We contrast the performance of surface-normal model trained from scratch (as in (Bansal et al., 2017)) in the second row with our model in the third row. We observe a significant 2% performance improvement. This means better surface normal estimation amounts to a better initialization for semantic segmentation, and that we have a robust representation that can be used for down-stream tasks.

**Can we improve semantic segmentation further?** Can we still improve the performance of a task when we start from a better initialization other than scratch? We contrast the performance of the methods in the third row (init) to the fourth row (improvement in one-iteration). We observe another significant 2.7% improvement in IoU. This conveys that we can indeed apply our insights even when starting from an initialization better than scratch. Finally, we observe that our approach has closed the gap between ImageNet (with class labels) pre-trained model and a self-supervised model to 3.6%.

## 4.2 Streaming Learning

We now demonstrate streaming learning for well studied fine-grained image classification in Section 4.2.1 where many years of research and domain knowledge (such as better loss functions (Azuri & Weinshall, 2020; Barz & Denzler, 2020), pre-trained models, or hyperparameter tuning) has helped in improving the results. Here we show that streaming learning can reach close to that performance in few days without using any of this knowledge. In these experiments, we randomly sample from 14M images of ImageNet-21K (Deng et al., 2009) without ground truth labels as the unlabeled dataset.

### 4.2.1 Fine-Grained Image Classification

We first describe our experimental setup and then study this task using: (1) **Flowers-102** (Nilsback & Zisserman, 2008) that has 10 labeled examples per class; (2) **CUB-200** (Welinder et al., 2010) that has 30 labeled examples per class; and (3) finally, we have also added analysis on a randomly sampled 20 examples per class from ImageNet-1k (Russakovsky et al., 2015) (which we termed as **TwentyI-1000**). We use the original validation set (Russakovsky et al., 2015) for this setup.

**Model:** We use the ResNet (He et al., 2016) model family as the hypothesis classes in Alg. 1, including ResNet-18, ResNet-34, ResNet-50, ResNext-50, and ResNext-101 (Xie et al., 2017). The models are ranked in an increasing order of model complexity. Model weights are randomly generated by He initialization (He et al., 2015) (a random gaussian distribution) unless otherwise specified. We show in Appendix A.4 that training deeper neural networks with few labeled examples is non-trivial.

**Learning $F$ from the labeled sample $S$:** Given the low-shot training set, we use the cross entropy loss to train the recognition model. We adopt the SGD optimizer with momentum 0.9 and a L2 weight decay of 0.0001. The initial learning rate is 0.1 for all experiments and other hyper-parameters (including number of iterations and learning rate decay) can be found in Appendix A.4.

| Semantic Segmentation on VOC-2012 | | | | | | | | | |
|---|---|---|---|---|---|---|---|---|---|
| | aero | bike | bird | boat | bottle | bus | car | cat | chair | cow |
| scratch-init | 62.3 | 26.8 | 41.4 | 34.9 | 44.8 | 72.2 | 59.5 | 56.0 | 16.2 | 49.9 |
| normals-init | 71.8 | 29.7 | 51.8 | 42.1 | 47.8 | 77.9 | 65.9 | 59.7 | 19.7 | 50.8 |
| normalsStream-init | 74.4 | 34.5 | 60.5 | 47.3 | 57.1 | 74.3 | 73.1 | 61.7 | 22.4 | 51.4 |
| +one-iteration | 82.2 | 35.1 | 62.0 | 47.4 | 62.1 | 76.6 | 74.1 | 62.7 | 23.9 | 49.9 |
| Bansal et al. (2017) | 79.0 | 33.5 | 69.4 | 51.7 | 66.8 | 79.3 | 75.8 | 72.4 | 25.1 | 57.8 |

| | table | dog | horse | mbike | person | plant | sheep | sofa | train | tv | bg | **IoU** ↑ |
|---|---|---|---|---|---|---|---|---|---|---|---|---|
| scratch-init | 45.0 | 49.7 | 53.3 | 63.6 | 65.4 | 26.5 | 46.9 | 37.6 | 57.0 | 40.4 | 85.2 | **49.3** |
| normals-init | 45.9 | 55.0 | 59.1 | 68.2 | 69.3 | 32.5 | 54.3 | 42.1 | 60.8 | 43.8 | 87.6 | **54.1** |
| normalsStream-init | 36.4 | 52.0 | 60.9 | 68.5 | 69.1 | 37.6 | 58.0 | 34.3 | 64.3 | 50.2 | 90.0 | **56.1** |
| +one-iteration | 47.0 | 55.5 | 58.0 | 74.9 | 73.9 | 40.1 | 56.4 | 43.6 | 65.4 | 52.8 | 90.9 | **58.8** |
| Bansal et al. (2017) | 52.0 | 65.8 | 68.2 | 71.2 | 74.0 | 44.1 | 63.7 | 43.4 | 69.3 | 56.4 | 91.1 | **62.4** |

Table 6: The goal of this experiment is to study two things: **(1) Can task-specific representations learned on unlabeled streams generalize to other tasks?** This allows us to study the robustness of our learned representations. We consider the target task of semantic segmentation and the source task of surface-normal estimation. Segmentation networks initialized with surface-normal networks already outperform random initialization (row2 vs row1), and further improve by 2% when initialized with stream-trained networks (row3). **(2) Can we still further improve the performance of a task when starting from an initialization better than scratch?** We then perform one additional iteration of stream learning (row4 vs row3), resulting in another 2.7% improvement, closing the gap between ImageNet pre-training to 3.6%.

**Learning $F'$ from $U$ with pseudo labels:** Once we learn $F$, we use it to generate labels on a set of randomly sampled images from *ImageNet-21K* dataset to get pseudo-labelled $U$. Then we randomly initialize a new model $F'$ as we do for $F$, then apply same network training for $F'$ on $U$.

**Finetuning $F'$ on labeled sample $S$:** After training $F'$ on the pseudo-labeled $U$, we finetune $F'$ on the original low-shot training set with the same training procedure and hyper-parameters. We use this finetuned model $F'$ for test set evaluation.

**Streaming Schedule and Model Selection:** We empirically observe that instead of training on entire unlabeled set $U$, we can slice up $U$ into a streaming collections $U_t$ for better performance. In these experiments, we use three iterations of our approach. We have 1M samples in $U_1$ (the same images as in *ImageNet-1K*), 3M samples in $U_2$, and 7M samples in $U_3$. We initialize the task using a ResNet-18 model (ResNet-18 gets competitive performance and requires less computational resources as shown in Table 10). We use a ResNext-50 model as $F'$ to train on $U_1$ and $U_2$, and a ResNext-101 model to train on $U_3$. These design decisions are based on empirical and pragmatic observations shown in Appendix A.5. Table 7 shows continuous improvement for various image-classification tasks at every iteration when using a few-labeled samples and training a model from scratch. We see similar trends for three different tasks. We are also able to bridge the gap between the popularly used pre-trained model (initialized using 1.2M labeled examples (Russakovsky et al., 2015)) and a model trained from scratch without any extra domain knowledge or dataset/task-specific assumption.

### 4.2.2 Why Streaming Learning?

We study different questions here to understand our system.

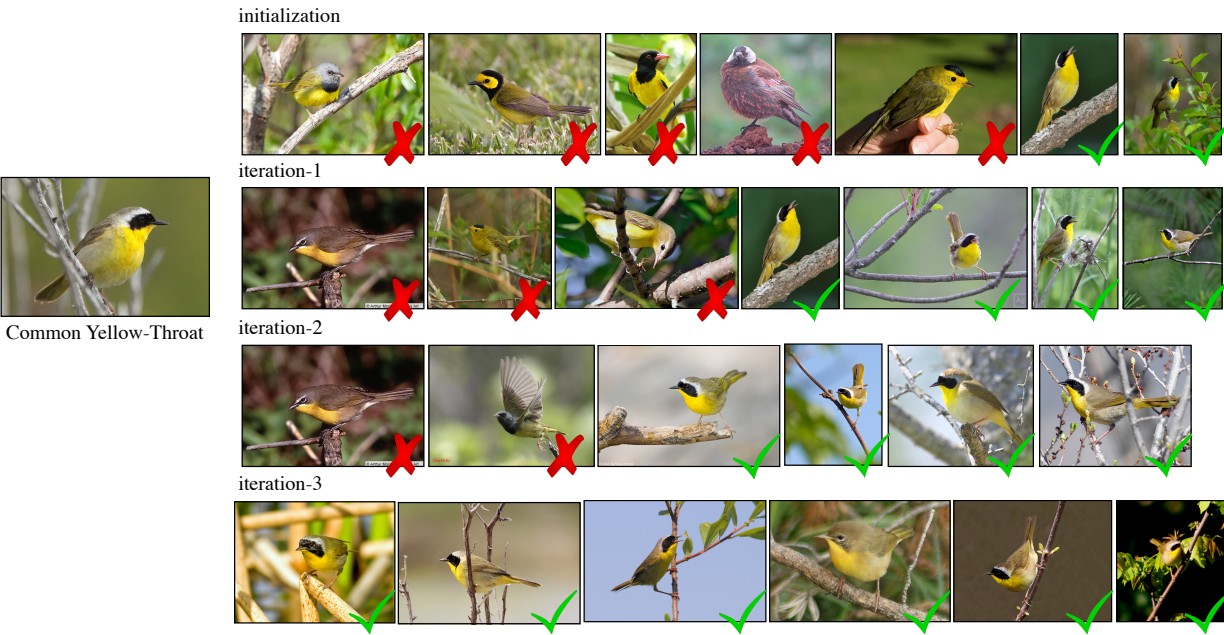

Figure 4: **Improvement in Recognizing Birds via Streaming Learning:** We qualitatively show improvement in recognizing a common yellow-throat (shown in left from CUB-200 dataset (Welinder et al., 2010)). At **initialization**, the trained model confuses common yellow-throat with hooded oriole, hooded warbler, wilson rbler, yellow-breasted chat, and other similar looking birds. We get rid of false-positives with every **iteration**. At the the end of the third iteration, there are no more false-positives.

| Continuously Improving Image Classification | | | | | | |
|---|---|---|---|---|---|---|
| **Task** | pre-trained | init | $U_1$ | $U_2$ | $U_3$ | ... |
| Flowers-102 | 89.12 | 45.56 | 54.19 | 65.25 | 72.79 | ... |
| CUB-200 | 75.29 | 44.03 | 53.73 | 57.11 | 66.10 | ... |
| TwentyI-1000 | 77.62 | 13.92 | 22.79 | 24.94 | 27.27 | ... |

Table 7: We continuously improve the performance for Flowers-102, CUB-200, and TwentyI-1000, as shown by top-1 accuracy for each iteration. We achieve a large performance improvement for each iteration for all the tasks. This is due to the combination of both increasing unlabeled dataset and model size. Without any supervision, we can bridge the gap between an ImageNet-1k pre-trained model and a model trained from scratch on Flowers-102 and CUB-200 dataset using a simple softmax loss.

**What if we fix the model size in the iterations?** We observe that using deeper model could lead to faster improvement of the performance. For the *TwentyI-1000* experiment in section 4.2.1, we perform an ablative study by only training a ResNet-18 model, as shown in Table 8. We could still see the accuracy improving with more unlabeled data, but increasing model capacity turns out to be more effective.

**What if we train without streaming?** Intuitively, more iterations with our algorithm should lead to an increased performance. We verify this hypothesis by conducting another ablative study on *TwentyI-1000* experiment in section 4.2.1. In Table 9, we compare the result the result of training with three iterations (sequentially trained on $U_1,U_2,U_3$) with that of a single iteration (that concatenated all three slices together). Training on streams is more effective because improved performance on previous slices translates to more accurate pseudo-labels on future slices.

**Cost of Experiments:** We now study the financial aspect of the streaming learning vs. single iteration via computing the cost in terms of time and money. We are given 11M unlabeled images and there are two

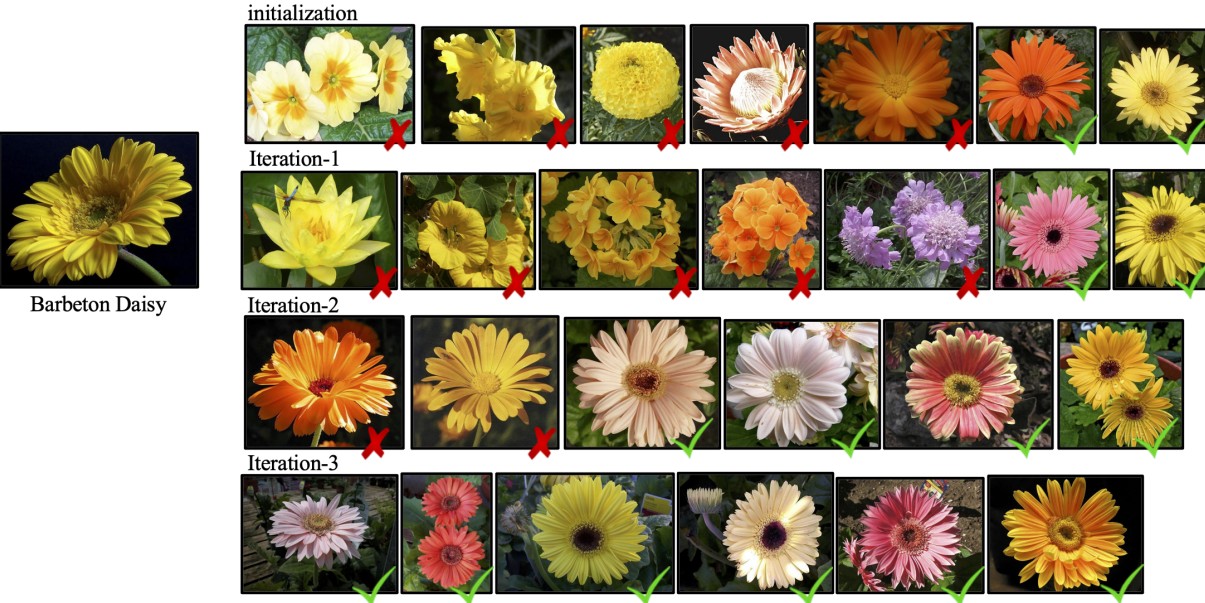

Figure 5: **Improvement in Recognizing Flowers via Streaming Learning:** We qualitatively show improvement in recognizing a barbeton daisy (shown in left from Flowers-102 dataset (Nilsback & Zisserman, 2008)). At **initialization**, the trained model confuses barbeton daisy with primula, water lily, daffodil, sweet william, and etc. With more iterations, the false positives become fewer.

| What if we use ResNet-18 for all experiments? | | | | | |
|---|---|---|---|---|---|
| **Model** | init | $U_1$ | $U_2$ | $U_3$ | ... |
| ResNet-18 only | 13.92 | 19.61 | 21.22 | 22.13 | ... |
| StreamLearning | 13.92 | **22.79** | **24.94** | **27.27** | ... |

Table 8: We show that the top-1 validation accuracy on TwentyI-1000 for our **StreamLearning** approach (row 2) for each iteration, which increases the model capacity from ResNet-18 (init) to ResNext-50 ($U_1$ and $U_2$) to ResNext-101 ($U_3$). With **ResNet-18 only** (row 1), the performance gain is much slower.

scenarios: (1) train without streaming ($U_1$) using 11M images and ResNext-101; and (2) train in streams ($U_1, U_2, U_3$) of {1M, 3M, 7M} images using ResNext-50 for $U_1$ and $U_2$, and ResNext101 for $U_3$. For $U_1$, we train $F'$ from scratch for 30 epochs. For $U_2$, we train $F'$ from scratch for 20 epochs. For $U_3$, we train $F'$ from scratch for 15 epochs. We could fit a batch of 256 images when using ResNext-50 on our 4 GPU machine. The average batch time is 0.39sec. Similarly, we could fit a batch of 128 images when using ResNext-101. The average batch time is 0.68sec. The total time for the first case (without streaming) is 486.96 hours (roughly 20 days). On the contrary, the total time for the streaming learning is 193.03 hours (roughly 8 days). Even if we get similar performance in two scenarios, we can get a working model in less than half time with streaming learning. A non-expert user can save roughly $1,470$ USD for a better performing model (60% reduction in cost), assuming they are charged 5 USD per hour of computation (on AWS).

## 5 Discussion

We present a simple and intuitive approach to semi-supervised learning on (potentially) infinite streams of unlabeled data. Our approach integrates insights from different bodies of work including self-training (Du et al., 2020; Wei et al., 2020), pseudo-labelling (Lee, 2013; Arazo et al., 2020; Iscen et al., 2019), continual/iterated learning (Kirby, 2001; Kirby et al., 2014; Thrun, 1996; 1998; Silver et al., 2013), and few-shot

| What if we train without streaming? | | | | | |
|---|---|---|---|---|---|
| **Model** | init | $U_1$ | $U_2$ | $U_3$ | ... |
| NoStreaming | 13.92 | **23.77** | – | – | – |
| StreamLearning | 13.92 | 22.79 | 24.94 | **27.27** | ... |

Table 9: We show that the top-1 validation accuracy on TwentyI-1000 for our **StreamLearning** approach (row 2) for each iteration, which increases the model capacity from ResNet-18 (init) to ResNext-50 ($U_1$ and $U_2$) to ResNext-101 ($U_3$). This result is compared to training with a **single iteration**, i.e, **NoStreaming**, that use ResNext-101 but with all the data.

learning (Li et al., 2019; Guo et al., 2020). We demonstrate a number of surprising conclusions: (1) Unlabeled *domain-agnostic* internet streams can be used to significantly improve models for specialized tasks and data domains, including surface normal prediction, semantic segmentation, and few-shot fine-grained image classification spanning diverse domains including medical, satellite, and agricultural imagery. In this work, we use unlabeled images from curated ImageNet-21k (Deng et al., 2009) and uncurated YFCC-100M (Thomee et al., 2016). We see more performance improvement as the unlabeled stream becomes more diverse. (2) Continual learning on streams can be initialized with very *impoverished models* trained (from scratch) on tens of labeled examples. This is in contrast with much work in semi-supervised learning that requires a good model for initialization. (3) Contrary to popular approaches in semi-supervised learning that make use of massive compute resources for storing and processing data, streaming learning requires *modest computational infrastructure* since it naturally breaks up massive datasets into slices that are manageable for processing. From this perspective, continual learning on streams can help democratize research and development for scalable, lifelong ML.

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

## A  Appendix

### A.1  Extreme-Task Differences

**Dataset:**  We randomly sample a 20-shot training set for each of the three datasets we present in the paper. For datasets without a test set, we curated a validation set by taking 10% of all samples from each category. Some of these datasets can be extremely different from natural images, and here we rank them in order of their similarity to natural images:

1. **CropDiseases** (Mohanty et al., 2016). Natural images but specialized in agricultural industry. It has 38 categories representing diseases for different types of crops.

2. **EuroSat** (Helber et al., 2019). Colored satellite images that are less similar to natural images as there is no perspective distortion. There is 10 categories representing the type of scenes, e.g., Forest, Highway, and etc.

3. **ISIC2018** (Codella et al., 2019). Medical images for lesion recognition. There is no perspective distortion and no longer contains natural scenes. There are 7 classes representing different lesion. Because the dataset is highly unbalanced, we create a balanced test set by randomly sampling 50 images from each class.

**Training details:** We use ResNet-18 only for all experiments on the 3 cross-domain datasets, in order to isolate the effect of data. We also only do one iteration of our approach, but still we see substantial improvement. The unlabeled set $U_1$ is still the unlabeled version of *Imagenet-1K* dataset. We intentionally do this in order to contrast with the performance by finetuning an *ImageNet-pretrained* model with is pretrained using the same images but with additional $1.2M$ labels. We use SGD optimizer with momentum 0.9 and a L2 weight decay of 0.0001.

**Learning $F$ from the labeled sample $S$:** For all these cross-domain few-shot datasets, we start with an initial learning rate of 0.1 while decaying it by a factor of 10 every 1500 epochs, and train for 4000 epochs.

**Learning $F'$ from $U$ with pseudo labels:** For $U_1$, we train $F'$ from scratch for 30 epochs starting from learning rate 0.1, and decay it to 0.01 after 25 epochs.

**Finetuning $F'$ on the labeled sample $S$:** We use the same training procedure when finetuning $F'$ on $S$.

## A.2 Surface Normal Estimation

**Model and hyperparameters:** We use the PixelNet model from (Bansal et al., 2017) for surface normal estimation. This network architecture consists of a VGG-16 style architecture (Simonyan & Zisserman, 2015) and a multi-layer perceptron (MLP) on top of it for pixel-level prediction. There are 13 convolutional layers and three fully connected (*fc*) layers in VGG-16 architecture. The first two *fcs* are transformed to convolutional filters following Long et al. (2015). We denote these transformed *fc* layers of VGG-16 as conv-6 and conv-7. All the layers are denoted as $\{1_1, 1_2, 2_1, 2_2, 3_1, 3_2, 3_3, 4_1, 4_2, 4_3, 5_1, 5_2, 5_3, 6, 7\}$. We use hypercolumn features from conv-$\{1_2, 2_2, 3_3, 4_3, 5_3, 7\}$. An MLP is used over hypercolumn features with 3-fully connected layers of size $4,096$ followed by ReLU (Krizhevsky et al., 2012) activations, where the last layer outputs predictions for 3 outputs ($n_x$, $n_y$, $n_z$) with a euclidean loss for regression. Finally, we use batch normalization (Ioffe & Szegedy, 2015) with each convolutional layer when training from scratch for faster convergence. More details about the architecture/model can be obtained from Bansal et al. (2017).

**Learning $F$ from the labeled sample $S$:** We use the above model, initialize it with a random gaussian distribution, and train it for NYU-v2 depth dataset (Silberman et al., 2012). The initial learning rate is set to 0.001, and it drops by a factor of 10 at step of $50,000$. The model is trained for $60,000$ iterations. We use all the parameters from Bansal et al. (2017), and have kept them fixed for our experiments to avoid any bias due to hyperparameter tuning.

**Learning $F'$ from $U$ with pseudo labels:** We use $F$ trained above to psuedo-label 1M images, and use it to learn a $F'$ initialized with random gaussian distribution and follows the same training procedure as $F$.

**Finetuning $F'$ on the labeled sample $S$:** Finally, we finetune $F'$ on $S$ for surface normal estimation. The initial learning rate is set to 0.001, and it drops by a factor of 10 at step of $50,000$.

We qualitatively show improvement in estimating surface normal from a single 2D image in Figure 6.

## A.3 Semantic Segmentation

We follow Bansal et al. (2017) for this experiment. The initial learning rate is set to 0.001, and it drops by a factor of 10 at step of $100,000$. The model is fine-tuned for $160,000$ iterations.

We follow the approach similar to surface normal estimation. We use the trained model on a million unlabeled images, and train a new model from scratch for segmentation. We used a batch-size of 5. The initial learning rate is also set to 0.001, and it drops by a factor of 10 at step of $250,000$. The model is trained for $300,000$ iterations. We then fine-tune this model using PASCAL dataset.

## A.4 Fine-Grained Image Classification

**Datasets:** We create few-shot versions of various popular image classification datasets for training. They are:

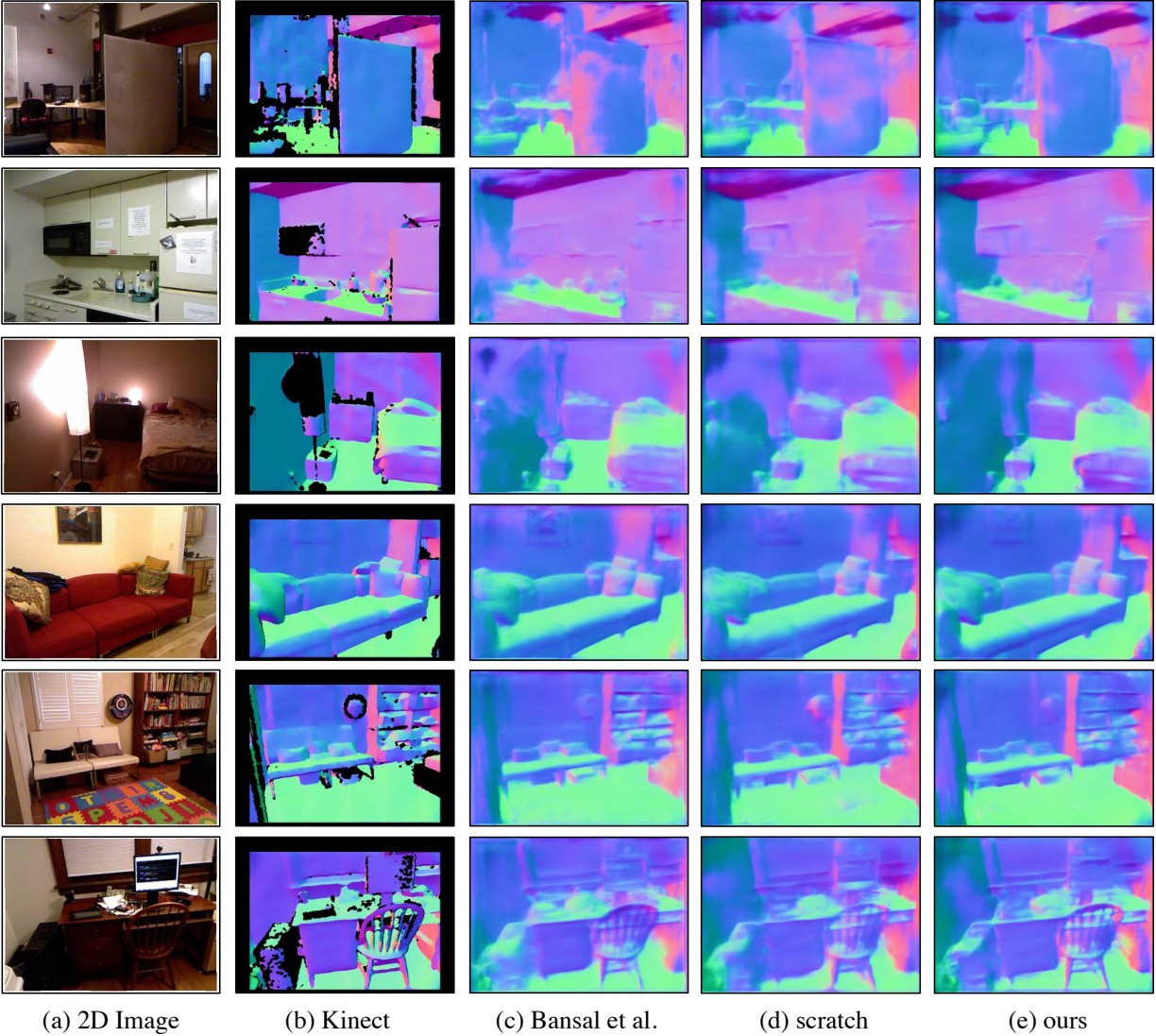

|   (a) 2D Image   |   (b) Kinect   |   (c) Bansal et al.   |   (d) scratch   |   (e) ours   |

Figure 6: **Surface Normal Estimation:** For a given single 2D image (shown in **(a)**), we contrast the performance of various models. Shown in **(c)** are the results from prior work Bansal et al. (2016; 2017) using a model pretrained with *ImageNet-1K* labels; **(d)** shows a model trained from scratch starting from random gaussian initialization; and finally **(e)** shows the result of our StreamLearning approach. The influence of unlabeled data can be gauged by improvements from **(d)** to **(e)**. By utilizing diverse unlabeled data, we can get better performance without any additional supervision. For reference, we also show ground truth normals from kinect in **(b)**.

1. **Flowers-102** (Nilsback & Zisserman, 2008). We train on the 10-shot versions of *Flowers* by randomly sampling 10 images per category from the training set. We report the top-1 accuracy on the test set for the 102 flower categories.

2. **CUB-200** (Welinder et al., 2010). We take 30 training examples per category from the Caltech UCSD Bird dataset and report the top-1 accuracy on the test set for the 200 birds categories.

3. **TwentyI-1000** (Russakovsky et al., 2015) (ILSVRC 2012 Challenge) with 1000 classes. Specially, we train on a 20-shot version of *ImageNet-1K*. We report the top-1 validation set accuracy on the 1000 classes as commonly done in literature (Xie et al., 2020b).

**Model and hyperparameters:** We experiment with the ResNet (He et al., 2016) model family, including ResNet-18, ResNet-34, ResNet-50, ResNext-50, and ResNext-101 (Xie et al., 2017). The models are ranked in an increasing order of model complexity. The initial model weights are randomly generated by He initialization (He et al., 2015), which is the PyTorch default initialization scheme. For all image classification experiments, we adopt the SGD optimizer with momentum 0.9 and a L2 weight decay of 0.0001. We use an initial learning rate of 0.1 for both finetuning on $S$ and training on $U$.

**Learning $F$ from the labeled sample $S$:** For **Flowers-102** (10-shot), we decay the learning rate by a factor of 10 every 100 epochs, and train for a total of 250 epochs. For **CUB-200** (30-shot), we decay the learning rate by a factor of 10 every 30 epochs, and train for 90 epochs. For **TwentyI-1000**, we decay the learning rate by a factor of 10 every 60 epochs, and train for a total of 150 epochs.

**Streaming Schedule:** We simulate an infinite unlabeled stream $U$ by randomly sampling images from *ImageNet-21K*. In practice, we slice the data into a streaming collections $U_t$. We have 1M samples in $U_1$, 3M samples in $U_2$, and 7M samples in $U_3$. We intentionally make $U_1$ the unlabeled version of *Imagenet-1K* dataset for comparison with other works that use the labeled version of *Imagenet-1K*.

**Model Selection:** We initialize the task using a ResNet-18 model because it achieved great generalization performance when training from scratch compared to deeper models and only costs modest number of parameters. We use a ResNext-50 model as $F'$ to train on $U_1$ and $U_2$, and a ResNext-101 model to train on $U_3$. These design decisions are based on empirical and pragmatic observations we provided in Appendix A.5.

**Learning $F'$ from $U$ with pseudo labels:** For $U_1$, we train $F'$ from scratch for 30 epochs starting from learning rate 0.1, and decay it to 0.01 after 25 epochs. For $U_2$, we train $F'$ from scratch for 20 epochs and decay the learning rate to 0.01 after 15 epochs. For $U_3$, we train $F'$ from scratch for 15 epochs and decay the learning rate to 0.01 after 10 epochs.

**Finetuning $F'$ on the labeled sample $S$:** We use the same training procedure when finetuning $F'$ on $S$.

## A.5 Ablative Analysis

We study different questions here to understand the working of our system.

**What is the performance of models trained from scratch?** We show performance of various models when trained from scratch in Table 10. We observe that training deeper neural networks from random initialization with few labeled examples is indeed non-trivial. Therefore, our approach helps deeper networks generalize better in such few shot settings.

**Why do we use ResNext-50 for $U_1$ and $U_2$?** We show in Table 11 that ResNext-50 outperforms ResNet-18 in first iteration to justify the model decision of our stream learning approach. Note that this is not saying ResNext-50 is the best performing model among all possible choices. For instance, ResNext-101 slightly outperforms ResNext-50 (around 1% improvement) on the first two iterations, but we still use ResNext-50 for $U_1$ and $U_2$ for pragmatic reasons (faster to train and save more memory). In practice, one can trade off generalization performance and training speed by select the most suitable model size just like what we did in this paper.

One-stage models trained from scratch

| Model | Flowers-102 | CUB-200 | TwentyI-1000 |
|---|---|---|---|
| Resnet-18 | **45.49** | **44.03** | **13.92** |
| Resnet-34 | 42.64 | **44.17** | **14.23** |
| Resnet-50 | 20.82 | 21.73 | 12.93 |
| Resnext-50 | 31.34 | 28.37 | 11.87 |
| Resnext-101 | 34.18 | 32.31 | 13.35 |

Table 10: We show performance of various models when trained from scratch. It is non-trivial to train a deep neural network with a few labeled examples as shown in this analysis. Despite increasing the capacity of the models and training them for longer, we do not observe any performance improvement.

Performance after $U_1$: ResNet-18 or ResNext-50?

| Model | CUB-200 | Flowers-102 | TwentyI-1000 |
|---|---|---|---|
| ResNet-18 | 51.35 | 47.50 | 19.61 |
| ResNext-50 | **53.73** | **54.19** | **22.79** |

Table 11: We show that the top-1 validation accuracy on all fine-grained classification datasets with our approach after the first iteration ($U_1$ with 1M unlabeled images) training with ResNet-18 or ResNext-50. We can see that ResNext-50 consistently outperforms ResNet-18 across all tasks.

