# OpenReview forum: "A Simple Baseline for Task-Agnostic Self-Training"
_TMLR — Withdrawn by Authors_

### Review · Reviewer_xAuv · 2023-07-25

**Summary Of Contributions:**

The authors study the task of semi-supervised learning and propose a curriculum learning approach where they iterate between self-training on a large unlabeled dataset and finetuning on a small labeled dataset. They show results on different image datasets, where some datasets are far away from the training distribution.

**Audience:**

No

**Broader Impact Concerns:**

The authors did not include a broader impact statement. I do not have ethical concerns in regards to this submission.

**Claims And Evidence:**

No

**Requested Changes:**


Overall, the proposed approach comes down to simple self-training which is very well-known to work well in self- or semi-supervised learning. The authors write that “Classical semi-supervised approaches achieve state-of-the-art results on various visual recognition tasks, especially image classification, but they are typically designed with expert knowledge of the task at hand such as task-specific data augmentation. However, these approaches do not generalize to novel tasks such as image segmentation and surface normal estimation.” But first, they show no evidence for the claim that regular SSL methods perform badly on image segmentation or surface estimation tasks. Second, SwAV and Noisy Student actually perform really well on the WILDS benchmark which is used to study transfer learning to tasks very different from regular image classification. Third, their proposed approach is actually very similar to the Noisy Student task, or many other SSL tasks which are based on self-learning, and thus, they do not propose something inherently novel. And, I do think that they contradict themselves by stating that regular SSL methods use hand-crafted augmentations and thus, do not generalize, but then the proposed method is in fact a typical SSL pseudo-labeling scenario.

The experiments are impossible to judge and many baselines are missing. Instead of using well-established benchmarks such as e.g. WILDS to track performance on a task that is dissimilar from the training task, the authors propose their own dataset. But now, the numbers cannot be judged, especially because the authors did not run all the relevant baselines we have for WILDS. Other experimental results are impossible to judge because the authors are using a model that is not common in literature so no literature values can be looked up to compare their values against. The values they report look too low for what they should be, at least in my intuition. For multiple experiments, the authors do not compare to any baseline, except for model training from scratch which is not sufficient. The reported results that they improve with each further iteration in Table 7 are far below the pretrained model and thus appear useless. The authors need to rework the experimental section and report results on standardized benchmarks.

From a methodological perspective, the proposed method looks very inefficient. As far as I understand, the authors discard the trained model in each iteration, after using it to finetune it on the labeled dataset and pseudo-label another portion of the unlabeled dataset U. And then they train a new model. It is not clear to me that this is even close to optimal. Does DINO perform worse on all the tasks the authors are interested in? How about Noisy Student? The authors need to work out the exact differences of their proposed method to a SotA algorithm such as DINO, SimCLR, SimSiam, BYOL or AdaMatch and then compare to those as well. In the end, they want to use a large unlabeled dataset for pretraining and then a smaller dataset for finetuning. There are many SSL methods which are designed to do exactly that which were not considered.


**Strengths And Weaknesses:**

## Detailed comments:

### Abstract:
“However, these approaches do not generalize to novel tasks such as image segmentation and surface normal estimation.” Unsupported claim, I have not seen evidence in the paper for it.

The intuition that the model progressively learns more complex features and benefits from more data as the learning goes is strongly related to curriculum learning and is not novel, e.g. [E].

[E] Bengio et al. “Curriculum Learning”

### Section 3:
“In this case, Step 1 may take an exorbitant amount of time to finish labeling on U.” Why don’t the authors just use entropy minimization [F,G]? There, the soft pseudo-labels are calculated on the fly and it actually works better than hard pseudo-labeling, at least in test-time adaptation settings.

[F] Grandvalet and Bengio: “Semi-supervised learning by entropy minimization”

[G] Wang et al. “Tent: Fully Test-time Adaptation by Entropy Minimization”

### Section 4.1.1:
Section 4.1.1. “In specific, we use SGD optimizer with momentum 0.9 and the default augmentation” -> please specify what the default augmentation is. Also, multiple grammar errors in this sentence.

It would be helpful to summarize what FixMatch does and how it works.

The results in this section are not comparable to the literature. In [A], Table 10, top1 accuracy on this dataset is above 95.6% for a ResNet50 trained with DINO. In [B], Table B.5., the supervised performance is 94.7% for a RN50. This result is for finetuning of a pretrained model though, so it is not comparable to training from scratch. The reported result here is 58.21% and even without using a pretrained backbone it looks low. In [C], Table 1, a ResNet50 gets a top1 accuracy of 85.8%, and this should be a setting equivalent to the authors’ “from scratch” setting. For comparison, a RN18 achieves 69.8% top1 accuracy on ImageNet with supervised ImageNet training (https://pytorch.org/vision/stable/models.html), while a RN50 achieves 76.1%. This is a gap of 6-7 p.p. and this is the range I would expect on Flowers as well. In contrast, the authors report a drop of around 30p.p. which seems excessive, and I think there might be an optimization issue. I think the authors should consider running their experiments using a ResNet50 architecture because the numbers will be more comparable to the literature then.
[A] Caron et al. “Emerging Properties in Self-Supervised Vision Transformers”
[B] Chen et al. “A Simple Framework for Contrastive Learning of Visual Representations”
[C] Zhai et al. “A Large-scale Study of Representation Learning with the Visual Task Adaptation Benchmark”

I could not find many papers reporting results on the CUB-200 dataset, but for example [D] reports accuracy numbers above 80% for most approaches, while the authors report numbers around 40-60% which are much lower.
[D] He et al. “Fine-grained Visual-textual Representation Learning”

For both of these settings, literature numbers must be cited. For this, the authors need to run their experiments on a model which is commonly used for these datasets.

### Section 4.1.2:
Here, the authors study tasks that are very different from the pretraining one. This is the setting where the WILDS benchmark should typically be used (https://github.com/p-lambda/wilds). WILDS contains tasks in medical imaging, satellite imaging, micro-biology microscopy etc, and is a standardized way to test generalization across very different tasks. While it is good that the proposed approach improves upon training from scratch on those newly introduced datasets, the numbers are impossible to judge, also because there is no comparison to other baselines. The authors should test their method on WILDS because this is the standardized benchmark to test exactly what they want to test.

### Section 4.1.3:
Here, the relevant benchmark is available on papers_with_code: https://paperswithcode.com/sota/surface-normals-estimation-on-nyu-depth-v2-1. The reported numbers for 11.25% are all around 60% top1 accuracy while the authors report 51.3% in Table 3 which is far below the state of the art. As the results currently stand, it is not possible to judge them. The only meaningful point of comparison the authors include is the baseline model trained from scratch, but they should compare to other models from the literature, especially those from the papers with code benchmark.

### Section 4.2.1:
“We empirically observe that instead of training on entire unlabeled set U, we can slice up U into a streaming collections Ut for better performance.” -> Is there empirical evidence for this reported anywhere in the paper?


Table 7: The dots at the end of the Table are not helpful and the authors should report the topline performance they get, preferably with a graph which shows the pretrained performance as a horizontal line and then plot the performance vs the iteration U_i. It is necessary to track what final number the proposed approach converges to. Right now, the best reported number for TwentyI-1000 is 27.27% which is far below the baseline result of 77.62%.


I believe that the fact that authors continuously improve performance with further training is trivial and is encountered in every single training run when training a machine learning model over multiple epochs.


### Section: Discussion:

“We demonstrate a number of surprising conclusions: (1) Unlabeled domain-agnostic internet streams can be used to significantly improve models for specialized tasks and data domains, including surface normal prediction, semantic segmentation, and few-shot fine-grained image classification spanning diverse domains including medical, satellite, and agricultural imagery.” -> This finding is not surprising but well-known, see the leaderboard of WILDS: https://wilds.stanford.edu/leaderboard/ . Here, Noisy Student and SwAV perform quite well and both use large amounts of unlabeled data.


### Typos:

“pseudo” is frequently misspelled.

---

### Review · Reviewer_5AAR · 2023-07-29

**Summary Of Contributions:**

The paper presents a simple method for conducting semi-supervised learning, with a focus on instances where the labeled data originates from a different distribution than the unlabeled samples (OOD). After initializing a small model using only supervised data, the method alternates between distilling the model using unsupervised data and fine-tuning it on the supervised set, expanding the model's capacity with each iteration. The conducted experiments evaluate the efficiency of this technique, specifically in the OOD case, on both global and local tasks.

**Audience:**

Yes

**Claims And Evidence:**

Yes

**Requested Changes:**

see above.
main ones are:
* compare with more recent ands suitable methods
* use SSL initialized models
* fuse multiple datastes
* explain why OOD is addressed by the method

**Strengths And Weaknesses:**

Strengths

(+) The task setup, which utilizes out-of-domain unsupervised data, is crucial since, in many cases, access to many samples, even unlabeled ones, is not available.

(+) It is impressive to observe that in tasks where ImageNet pre-training fails to generate useful initialization, the proposed method consistently improves "from-scratch" performance.

Weaknesses

(-) My primary concern is that the methods used for comparison are neither the most recent nor the most suitable baselines. For more recent examples, please refer to the list of reference papers. In particular, the role of recent “foundation” models like SAM [7] aim to perform well with little or even no training data.

(-) The other main concern, is that i don’t understand what would make the method shine in the OOD case? this doesn’t seem to be directly addressed.

(-) The comparison with FixMatch is intended to demonstrate that out-of-domain data is better utilized in the proposed approach than in FixMatch. A few questions:

- In the method implementation, was there any model growth used?
- Was thresholding employed?
- What hypothesis leads to the belief that the proposed recipe is better suited for handling out-of-distribution data over semi-supervised methods like FixMatch? Do the authors think that using both labeled and unlabeled data simultaneously is harmful, and is that "fixed" by training on the unlabeled set and refining on the labeled set separately? If so, why not perform ablation using a more direct comparison than FixMatch, which includes other differences?
- The results indicating that FixMatch diminishes performance compared to training from scratch raise concerns. Perhaps parameter tuning is necessary? In FixMatch, the parameters used for ImageNet differed from those for other datasets, so it's not unreasonable to think that improved parameters might be needed.
- I would be interested in seeing how the proposed method compares to FixMatch in in-domain.
- Lastly, there are newer methods that outperform FixMatch. It would be beneficial to see how the proposed method compares with these.

(-) In the "Extreme-Task Differences" experiment, a more natural comparison than supervised ImageNet might be a self-supervised pre-trained network, such as [1], or its follow-ups. These can be tested with pre-training on the corresponding dataset. It's worth noting that self-supervised pretraining for initialization has proven effective in conjunction with label propagation [2].

The third row indicates that using a supervised pre-trained ImageNet model is considerably better than employing the distillation-based training proposed here. However, methods like SimCLRv2 have shown to match the supervised pre-training performance when fine-tuned.

(-) This observation highlights a limitation of the proposed method: as it continuously modifies the underlying model, it can't take advantage of strong pre-trained initializations (though each model could potentially be trained separately). Still, I would be interested in seeing the impact of, say, using a pre-trained model in the final distillation cycle.

Minor Points

(-) In light of significant works like [1,3,4], phrases such as "This suggests that we should not limit ourselves to pre-trained classification models that have access to large labeled datasets" appear somewhat dated.

(-) In Figure 4, why not use the same images across all rows to demonstrate how the model improves its predictions?

(-) why not try to mix multiple datasets for the unsupervised distillation stage?

References:

[1] A Simple Framework for Contrastive Learning of Visual Representations

[2] Contrast to Divide: Self-Supervised Pre-Training for Learning with Noisy Labels

[3] Momentum Contrast for Unsupervised Visual Representation Learning

[4] Barlow Twins: Self-Supervised Learning via Redundancy Reduction

[5] DreamTeacher: Pretraining Image Backbones with Deep Generative Models

[6] Masked Autoencoders are Scalable Vision Learners

[7] Segment Anything

---

> ### Comment · Action_Editors · 2023-09-06
> **official comment**
>
> Dear Reviewer 5AAR,
>
> Thanks for the very detailed comments and suggestions. Unfortunately, for some reason, we did not have the rebuttal and updated manuscript from the authors. As the deadline of this paper has been passed, could you please submit the final recommendation to this paper? Thanks!
>
> best,
>
> AE

---

### Review · Reviewer_oaDg · 2023-08-06

**Summary Of Contributions:**

The authors introduce a task-agnostic self-training strategy that contrasts with conventional methods dependent on task-specific knowledge and data augmentation.

**Audience:**

Yes

**Claims And Evidence:**

Yes

**Requested Changes:**

1. suggest the authors emphasize the unique benefits of their approach within established pretraining and fine-tuning methodologies.
2. suggest authors to enhance transparency by adding model size and training iterations in their performance comparisons.

**Strengths And Weaknesses:**

Strengths:

1. The method is task-agnostic and can effective utilize data that may vastly differ from the specific task data. This versatility holds promise for broad applicability across a range of tasks.

2. The authors considered wide scenarios to evaluate the method.

Weaknesses:

1. The proposed method shows lower performance than pretraining and fine-tuning workflow. Highlighting how this approach enhances upon such a framework, and articulating its specific advantages, would strengthen the method's rationale and appeal.
2. To provide readers with a clearer understanding and ensure equitable comparisons, inclusion of specific details like model size and iteration counts within performance comparison tables would be valuable, especially given the method's reliance on increased model size and iterative training on unlabeled streams.

---

### Comment · Action_Editors · 2023-08-22
**Hi**

Dear Authors,

Maybe just busy with NeurPIS rebuttal. Could you please reply the questions from each reviewer? Thanks!

best,

AE

---

### Note · Authors · 2023-09-07

**Comment:**

Dear Editor and Reviewers,

We thank you for the insightful comments and reviews about our work. Reviewers have suggested important baselines to be incorporated in the manuscript. It would require us time to run the suggested experiments. We have, therefore, decided to withdraw our submission at this point in time.

Thanks,
Authors

**Withdrawal Confirmation:**

I have read and agree with the venue's withdrawal policy on behalf of myself and my co-authors.